# Effects of Dance Movement Therapy on Adult Patients with Autism Spectrum Disorder: A Randomized Controlled Trial

**DOI:** 10.3390/bs8070061

**Published:** 2018-06-29

**Authors:** Anna Mastrominico, Thomas Fuchs, Elizabeth Manders, Lena Steffinger, Dusan Hirjak, Maik Sieber, Elisabeth Thomas, Anja Holzinger, Ariane Konrad, Nina Bopp, Sabine C. Koch

**Affiliations:** 1Department of Therapy Sciences, SRH University Heidelberg, 69123 Heidelberg, Germany; skoch@srh.de; 2Department of Psychiatry, University of Heidelberg, 69115 Heidelberg, Germany; thomas.fuchs@med.uni-heidelberg.de; 3Creative Arts Therapies, Drexel University, Philadelphia, PA 19102, USA; ejm94@drexel.edu; 4University College, University of Hamburg, 20146 Hamburg, Germany; lena.steffinger@uni-hamburg.de; 5Department of Psychiatry and Psychotherapy, Central Institute of Mental Health, Medical Faculty Mannheim, 68159 Mannheim, Germany; dusan.hirjak@zi-mannheim.de; 6SALO AG, Ludwigspl. 9A, 67059 Ludwigshafen, Germany; maiksieber@salo-ag.de (M.S.); thomas.elisabeth@web.de (E.T.); 7Forensic Department, Psychiatrisches Zentrum Nordbaden, Heidelberger Str. 1a, 69168 Wiesloch, Germany; anja.holzinger@t-online.de; 8Psychosomatic Department, Fürst Stirum Klinik Bruchsal, Gutleutstraße 1/14, 76646 Bruchsal, Germany; info@philodanza.de; 9Psychiatry Department, Johannes-Diakonie Mosbach, Neckarburkener Str. 2, 74821 Mosbach, Germany; tanztherapienb@aol.com; 10Research Institute for Creative Arts Therapies (RIArT), Alanus University Alfter, 53347 Alfter, Germany; sabine.koch@alanus.edu

**Keywords:** dance movement therapy, DMT, autism spectrum disorder (ASD), randomized controlled trial, embodiment, empathy, mirroring

## Abstract

This study examines the effects of dance movement therapy (DMT) on empathy for adults with autism spectrum disorder (ASD). DMT based on the embodiment approach offers body-centered interventions, such as mirroring techniques, to address the needs of ASD patients. Accordingly, findings of a feasibility study suggest that DMT may be an effective approach for clients on the ASD spectrum. The present study is a randomized controlled trial that was conducted as a multicenter study within the framework of the EU-funded research project TESIS (Toward an Embodied Science of Intersubjectivity), and employed a two-factorial between-subject design. The treatment group (*n* = 35) participated in a 10-week manualized DMT intervention, whereas the control group (*n* = 22) received treatment only after a waiting period. Empathy, measured with the *Cognitive and Emotional Empathy Questionnaire* (*CEEQ*), was the main variable of interest, analyzed by a repeated measures analysis of variance. In order to also include incomplete data cases, we used the expectation-maximization algorithm for missing data estimation. Results suggest no significant changes in overall empathy between groups. We discuss the results and limitations, as well as future research options.

## 1. Introduction

There has been an increasing interest in the use of dance movement therapy (DMT) in autism spectrum disorder [1]. Autism spectrum disorder (ASD) is a developmental disorder, marked by impairments in reciprocal social interactions and patterns of communication as well as by a restrictive, stereotyped, and repetitive repertoire of interests and activities (as defined in the *International Classification of Diseases-10*, ICD-10, by the World Health Organization, WHO). These impairments affect a broad range of daily life activities and the functioning of an individual in a variety of different situations. In the *Diagnostic and Statistical Manual of Mental Disorders* version 5 [2], the previously differentiated underlying categories of autism are subsumed under the broader term ASD, accounting for frequent overlaps and difficulties in differentiating the symptomatology of different subtypes of individuals diagnosed with this disorder, and also for the huge interindividual scope of the disorder. With respect to genetics, ASD shows a high heritability rate [3] and epidemiology prevalence rates are estimated to range between 0.6 and 0.7% [4]. Deficits in social cognition seem to constitute one of the central impairments in ASD (e.g., [5]).

Individuals with ASD also frequently have sensory and/or motor challenges which may interfere with emotional regulation, social engagement, and the development of social competency [6,7,8]. Sensory differences can include heightened or reduced sensitivity to information from all of the senses at times overwhelming the individual or causing other emotional reactions [8]. Motor differences include coordination, fine and gross motor challenges, and postural difficulties [6,8], and managing these may reduce the cognitive capacity available for social interactions [6]. These motor challenges may also interfere with the ability to coordinate one’s actions in time with a social partner as is necessary for successful joint attention and interpersonal synchrony [6,7].

A widely used approach to explaining the nature of social cognition is *Theory of Mind* (ToM) which includes several different variations and hybrid theories [9]. ToM refers to the capacity of attributing mental states to self and others and is crucial for social cognition. Missing ToM capacities constitute one of the primary explanations for deficits in perspective taking and empathy in individuals with ASD. Difficulties with ToM tests commonly used in ASD studies, however, are neither unique nor universal across individuals with ASD, leading some researchers to include more complex explanations and combinations of theories under ToM [10]. As reported in a study by Lombardo, Barnes, Wheelwright, and Baron-Cohen [11], individuals with ASD suffer from impairments in empathy as well as in self-referencing cognition. As an alternative to the theory of mind approach, interaction theory offers a more comprehensive way to explain symptoms in ASD [12]. According to interaction theory, humans are endowed with the capacity of intersubjective understanding. This capacity exists from birth on, develops in the early stages of life, and can be divided in primary intersubjectivity (i.e., early developed embodied understanding of others’ intentions and feelings in interactions on the level of perceptual experience; [13]) and secondary intersubjectivity (i.e., expanding this understanding to objects or events entering the interaction; [14]). For individuals with ASD, intersubjective understanding is affected [12,15]. Specifically, Mundy, Gwaltney, and Henderson [16] propose that for ASD patients an early impairment of the processing of self- and other-related sensory information exists. The embodiment approach, which serves as an alternative approach underscored by a phenomenological perspective, argues that cognition and emotion are grounded in bodily states. This approach supports treatment with embodied therapy methods such as dance movement therapy (DMT), because they can foster embodied experiences through body-based interventions [1,12,17,18].

The construct of empathy can be described as being comprised of different components including emotions (i.e., emotional responses to an affective state of another person) and cognitions (i.e., understanding the feelings of others, using ToM; [19]). Besides these two common aspects of empathy, motor empathy and kinesthetic empathy have been suggested as forms of empathy based on bodily movement or embodied responses. *Dance* movement therapists have long used kinesthetic empathy to describe the experience of connecting to the client’s experience through one’s own bodily sensations by attending to one’s own internal reactions and sensations when mirroring or observing the other’s movements [20] (see Box 1 for an example on the haptic level). Motor empathy is defined as an automatic sensation or reaction to the observation of another persons’ actions [21] through the activation of mirror neurons [22]. Research in neurosciences has demonstrated that an individuals’ mirror neurons [22] are activated both, when the individual performs a goal-oriented action or observes this action in another individual. The activation of these mirror neurons during the observation of another’s behavior is expected to help an individual understand and interpret the other’s behavior and its connected mental states [23]. A connection between mirror neurons and empathy seems plausible as empathic understanding between different individuals is proposed to be essentially influenced by a mechanism involving large networks of mirror neurons [24]. Blair [21] relates that individuals with ASD particularly show impairments in cognitive and motor empathy.

Box 1Example of impaired motor empathy in a child with autism spectrum disorder.Example of Paul, a three-year-old boy diagnosed with Autism Spectrum Disorder, treated with one-on-one dance movement therapy in an outpatient day-care center for autistic children:Paul profited much from dance movement therapy. He was one of the children who spoke their first word in a movement therapy session. When one day, author SK worked in a ball pit with Paul, the boy jumped down into the ball pit from the wall bars (a wooden tower/play structure) and fell with his head and upper body into the therapist’s stomach. SK, the therapist, shrank in pain, curling herself up with a painful sound. The boy did not react at all, not the slightest twitch. He lay there and was seemingly unaware of the meaning of the shrinking body underneath his head/upper body. This was a moment of realization in the therapist about how deep the social impairment reached into the sensory level, in the case of Paul. This realization was quite strong, because of the interaction on the skin-to-skin, muscle-tension-flow to muscle-tension-flow, haptic–tactile level.One needs, however, be aware of the fact that in autism reactions are sometimes extremely time-delayed, but still contingent. The session with Paul continued for another 15–20 min, yet in this time no reactions to the described event were observed, nor in the next session a few days later.

Autism spectrum disorders manifest as a variety of symptoms, which vary in their degree and overall importance depending on the individual. Treatments commonly include therapies or interventions which focus on the most prevalent symptoms, namely difficulties in social interactions and verbal as well as non-verbal communication, combined with the reduction and/or socially adaptive modifications of patterns of restricted or repetitive interests and activities [25]. Given the range of presentations of the symptoms and inconsistent responsiveness to interventions by individuals [26], it is important to investigate all potentially effective interventions for this population, among them dance movement therapy (DMT). In DMT, movement is used as a psychotherapeutic tool to foster the emotional, cognitive, social, and physical integration of the individual. This includes work on the body level to increase self-recognition, self–other distinction, self-awareness, social interaction and communication, and to decrease repetitive and dysfunctional behavior (e.g., self-harming behavior), taking into account the close relation of physical well-being, emotions, and cognition (increasing the emotional awareness and the ability to language the experience). Devereaux [25] underlines the unique possibilities offered by DMT, including work and reflection on the non-verbal level with opportunities to explore sensory and motor experiences and develop bodily awareness and expressivity in movement. Given the potential of sensory and motor challenges to impact individuals’ availability to engage socially [6,7,8], the facilitation of multiple ways of connecting to sensory and motor experiences including moving in and out of interpersonal synchrony within the context of an interaction in DMT, may contribute to its therapeutic success in treating individuals with ASD [27].

McGarry and Russo [28] assume that an enhancement of empathy for others through the mirroring experience in dance movement interventions is connected to an enhanced use of the mirror neuron system (MNS; [22,29], for further evidence see Winkielman, McIntosh, and Oberman [30]). Patient groups with deficits in the MNS may benefit from these interventions, because the engagement of individuals with ASD in a mirroring intervention is proposed to lead to an enhanced capacity to express oneself due to an enhancement of the mirror neuron system’s activity. One related technique used in DMT is mirroring. As proposed by Sandel, “mirroring is often the first step in establishing empathic connections, particularly with patients who are unresponsive to other modes of interpersonal exchange” [31] (p. 100). Generally, the technique of mirroring is used in DMT, to facilitate the development of the therapeutic relationship through reflections of body rhythms, movement patterns, and vocalization [32]. This can be considered as an important goal in the therapy for individuals with ASD, taking into account the difficulties in reciprocal interaction and communication of the persons affected. Importantly, the most decisive aspect in mirroring is the reflection of the quality of the movement rather than mechanically copy its precise shape. Eberhard-Kaechele [33] provides a detailed taxonomy of different mirroring modalities, developing from simple to more complex, which can be used to categorize the developmental stage of mirroring in movement of a client. Koch and colleagues [1] applied this concept in their feasibility study in order to observe and assess the developmental states of the participants over the time of the study. Following this notion, only in the very last modality, which develops between the ages of four and six, the capability of attributing mental states to oneself and others (ToM) is reached with certainty.

A DMT intervention such as mirroring offers a body focused therapy to clients with ASD and may improve empathy in persons with autism [28]. Koch [34] described the development of a DMT intervention based on embodiment for ASD including different mirroring exercises. The treatment manual by Koch and colleagues was used in the present study. The employment of a manualized form of DMT has already been reported beneficial for this population [35]. However, so far only a few evidence-based studies exist on the effects of DMT on individuals with ASD [36,37,38]. Some of the existing studies already employed quantitative research designs (e.g., [36,37]) or focused on the benefits of DMT for specific cluster of patients such as children (e.g., [38]). Adler’s classic video “Looking for me” [39] depicts the process of DMT with a three- and a five-year-old girl with ASD, illustrating the use and potential of DMT. While the use of DMT for children with ASD is frequently mentioned as a promising therapeutic approach [25], findings up to 2000 came exclusively from case-studies, and evidence for its application with adolescents or adults is still rudimentary [36].

With regards to adults, the findings of a feasibility study by Koch et al. [1] suggest that DMT based on therapeutic mirroring can indeed be an effective approach in therapy for adolescent and adult patients. In their study, participants received DMT for seven weeks, based on the same manualized intervention as the study at hand. After the intervention, participants in the treatment condition showed significantly increased values in psychological well-being, body awareness, self-other distinction and social skills compared to their peers. The effect of empathy was the only one that turned out to be nonsignificant, but the corresponding variable still showed a considerable and potentially clinical significant increase compared to the control group and baseline levels. Since the study was conducted as a feasibility study, the authors stated that additional studies with randomized designs were needed, possibly replacing the empathy measure with a more body-related scale for a higher change sensitivity. For this reason, the present study was conducted within the TESIS-project. Providing further evidence on the effectiveness of DMT in ASD patients, Hildebrandt et al. [36] found a trend of DMT reducing negative affect in ASD patients within the same sample of the TESIS project. Interestingly, the negative affect in individuals with autism was significantly higher than the negative affect in individuals with schizophrenia that were also part of the TESIS study, in a separate but parallel multicenter RCT [36], where DMT was also found to reduce negative affect significantly (measured with the SANS [40]). Finally, Koehne et al. [37] recently conducted a study investigating the effectiveness of a DMT intervention on emotional inference and empathic feelings of individuals with high-functioning ASD. Their results showed an improvement for emotion inference but not for empathic feelings [37]. Overall, reported results indicate that DMT may indeed have an effect on ASD patients by reducing some symptoms as well as reinforcing their existing positive resources in the domains of synchronization and spontaneous mirroring (e.g., Oberman, Winkielman, and Ramachandran [41]). 

Our randomized controlled trial (RCT) on the effectiveness of DMT for ASD patients was conducted within the framework of the EU-funded research project *Towards an Embodied Science of Intersubjectivity* (TESIS; 2011–2015). We build upon an unpublished Master’s thesis [42] in the TESIS-project, which had indicated positive effects of DMT for individuals with ASD on empathy. The present study performs a reanalysis of the data, including incomplete cases within the sample and further accounting for the results of the feasibility study [1]. The primary purpose of this study was to find out, whether a specific manualized DMT intervention has a positive effect on the self-reported empathy of young adults with ASD.

## 2. Materials and Methods

The present study was conducted using a two-factorial design with the factors *group* (treatment group versus control group) and *time* (before and after a 10-session manualized DMT intervention). The study design had been approved by the ethics committee of the medical faculty of the University of Heidelberg (Germany).

### 2.1. Sample

Seventy-three individuals with autism spectrum disorder participated in three different institutions in three cities of Southern Germany. After excluding eighteen participants due to completely missing values, fifty-seven participants remained for further analyses. Forty-four of the patients were male and twelve female. Ages ranged from 14 to 52 years (*M* = 22.5; *SD* = 8.52). Most participants were German, living without a partner and had no children. 26% reported taking medication and/or taking part in other therapies (e.g., physiotherapy, occupational therapy, social skills training). 26% reported comorbid psychiatric disorders (e.g., depression). See Table 1 for more details on the demographic data.

Recruitment of participants was accomplished through the institutions conducting the therapy sessions. Participants received a 15-Euros expense allowance after they had completed both testing sessions. Before being admitted to the study, all participants signed the informed consent form and had to meet the following inclusion criteria: German as a first language, age between 14 and 65, and a diagnosed ASD (ICD-10: F84.0, F84.1, F84.5, F84.9). Exclusion criteria were lack of consent given for the participation in the study, a present acute or transient psychotic disorder, any addictive disorder, mental retardation (IQ < 70) as well as severe neurological or internal diseases that would affect the participants’ ability to move. Participants did not pursue other therapy during this study except to continue any existing individual therapies (see also Table 1).

### 2.2. Procedure

The present study was a project of the Heidelberg node of the EU funded Marie Curie initial training network TESIS treating *Embodied Intersubjectivity in Schizophrenia and Autism* (Subproject Heidelberg Node). The experimental sessions for the present study took place in three different institutions and six different therapy groups. The number of participants per group ranged from five to ten. Before the first therapy session, participants completed a pretest consisting of questionnaires (including the measures analyzed for the study at hand) and a variety of diagnostic observational measures such as the SANS-scale for negative symptoms. Then, participants were randomly assigned to either the treatment group or the control group. Directly following the group assignment, the treatment group participated in the first DMT session, whereas the control group was placed on a wait-list and received no intervention or alternative treatment. They were to receive the same treatment after they had completed their follow-up testing. After the treatment group had completed a total of ten DMT sessions, they took the posttest consisting of the same questionnaires as used for the pretest plus some additional questions on the treatment. A follow-up was administered after six months, but did not provide enough data due to substantial drop out. Consequently, the follow-up data were not included into the statistical analysis.

The DMT intervention took place on a weekly basis in the cooperating institutions with a duration of 60 min. The intervention followed a manualized structure of DMT and was held by different female dance movement therapists, depending on the institutions where the sessions took place. Prior to the first session, both therapists received training on how to work with the manual. They were additionally supported by two cotherapists, students of the dance movement therapy program at SRH University Heidelberg or students majoring in Psychology at the University of Heidelberg, who had previously been instructed on the content of each therapy session. The whole intervention closely followed the structure described in the manual. Every dance movement session was built from the same elements, including a warm up, two different mirroring exercises, and a final verbal reflection part as described in Koch et al. [1], and Hildebrandt et al. [36]. Only this last part included substantial verbal intervention, whereas the first three parts were based on movement interventions. The following paragraphs briefly describe the elements within each session.

*Warm-up (~10 min).* The therapist facilitated the beginning of each session with a Chace-circle warm-up: Participants are in a circle while the therapist picks up movement elements of one participant after the other and asks the group to perform them together and change them or adapt them to their own movement. In this way, the circle also served as a starting point to experiment with elements of mirroring. The Chace circle creates an opportunity for each participant to experience others’ perception of him- or herself as a moving individual. This facilitates becoming part of the group while also establishing a relationship with the therapist [32].

*Dyadic movement (15–20 min).* After the warm-up, the group split up into dyads, which consisted of either a participant working with a therapist or cotherapist, or of two participants working together. Three different pieces of music varying from faster to slower rhythms were played. During the first piece of music, participant A moved with participant B mirroring the other’s movements. The task of the mirroring person was to react spontaneously and in an empathic manner to the movements observed. To do this, they were instructed to follow the first person, and were told that they did not need to always make the same size or exact shapes as the leader’s movement, but rather to try and capture the feeling or quality of the movements as had been modeled in the warm-up. For the second piece of music, the roles switched so that participant A now mirrored the movements executed by participant B. For the third piece of music, both participants were instructed to dance as they wished, but to still remain in contact with each other and relate to their partner’s movements. It was predicted that this intervention might encourage the participants to experience being mirrored and to explore their individual movements [31] in the secure atmosphere of the session.

*Baum Circle (~20 min).* In Section 3, participants again performed movements in a circle, this time choosing their own music and following the method of the Baum Circle [43]. In each session, up to five participants could volunteer to lead the group in a movement improvisation. One participant would start to improvise free movements and the rest of the group then followed the participant just as they had done in the dyads before. In all groups, the therapist or a cotherapist modeled this way of authentic self-expression to a piece of music in the very first session, followed in movement by all of the participants and the therapist. Participants’ task at this point was to take the movement quality and their emotional content into their own movements. As before, this intervention, through mirroring, serves to facilitate rapport and empathy and the feeling of being accepted and valued by the group [43].

*Verbal processing (10–15 min).* The Baum Circle was followed by time to discuss and reflect on the feelings and thoughts that have emerged during the session. The participants who previously improvised as a group now had the opportunity to discuss the movements they performed, how they felt during the improvisation, and how they perceived the mirroring by the other group members. Eventually, all participants were encouraged to verbalize the feelings and perceptions they experienced during the mirroring exercises in order to increase awareness of any outcomes experienced at the motor level and bring them to a cognitive level of processing. The cotherapists modeled the expression of emotions and thoughts about their experiences to facilitate the participants’ reflection processes. The verbal processing and feedback after the movement intervention was intended to strengthen body awareness, self-awareness, and self-other awareness as well as empathy, social skills [1] and the connection of sensorimotor and emotional-cognitive levels of experiencing and reflecting. This was anticipated to help participants connect their experiences of motor empathy during the earlier parts of the session to a more general understanding of the other’s perspective and experiences. Participants also completed brief questionnaires on their experience of the session and their connection with their partner, which may have further supported this process.

### 2.3. Measures

The measures used to analyze empathy were part of a battery of questionnaires employed at both testing sessions, pretest and posttest. Demographic data were assessed along with the questionnaires. All measures were in German. The measures analyzed in the study at hand are described in detail below.

The construct of empathy was measured with two different questionnaires. The Cognitive and Emotional Empathy Questionnaire (CEEQ, [44]) assesses cognitive and emotional aspects of empathy with 30 items. The scale *Cognitive Empathy* was divided into the subscales *Mental State Perception* and *Perspective Taking*, assessing the capability to identify and understand others’ emotions, feelings, and perspectives. The scale *Emotional Empathy* with its subscales *Mirroring* and *Empathic concern* assesses emotional reactions to the affective states of others [44]. All scales were answered on a five-point Likert scale, ranging from *0* (“Does not apply at all”) to *4* (“Applies exactly.”). Sample items for *Cognitive Empathy* were “I am not good at telling a real smile from a fake smile” (subscale *Mental State Perception*), and “When interacting with people, I am good at taking their perspective” (subscale *Perspective Taking*; inverted item). For *Emotional Empathy*, sample items were “Hearing other people laugh makes me to want laugh, too” (subscale *Mirroring*) and “I am often concerned about the feelings of people I like” (subscale *Empathic Concern*). Given that no psychometric measures were available for the German version of those scales, we referred to the satisfactory reliability and validity of the English version reported by the authors. The internal consistency as measured by Cronbach’s α of the subscales ranged from 0.69 to 0.83, and thus suggested that the aggregation of the subscales to the scales *Cognitive Empathy* and *Emotional Empathy* was appropriate.

In addition to the CEEQ, a second measure was employed to further assess empathy. The subscale *Empathic Concern* of the Interpersonal Reactivity Index (IRI, [45]) measures the disposition to care about the feelings and needs of other people. This scale is available as part of the SPF-E (Saarbrücker Persönlichkeitsfragebogen zur Messung von Empathie) in a German translation [46] and represents a commonly used scale to assess empathy in adults. It was selected because this aspect of empathy was not sufficiently covered by the other measure. The subscale *Empathic Concern* consists of four items, answered on a five-point scale ranging from *1* (“Does not describe me well.”) to *5* (“Describes me very well.”). A sample item was “I often have tender, concerned feelings for people less fortunate than me”. The author reports good validity and reliability with a Cronbach’s alpha for the subscale of α = 0.71 [46].

Regarding the validation of the questionnaires with ASD participants, the CEEQ was constructed with clinical populations (including ASD) in mind, but is too new to have been validated with the population. It, however seems intuitively suited, because of its many nonverbal components, including “mirroring”. The IRI/SPF-E has been used with samples of the autism spectrum [1], but we employed only the “Empathic Concern” subscale. The whole battery of questionnaires also included the Boundary Orientation between Self and Other Scale (GO), the Body Self-Efficacy Scale (BSE), and the Embodied Intersubjectivity Scale (EIS; see Appendix A) which were developed by our group particularly for the populations of autism and schizophrenia, and were pretested in our feasibility study [1]. As the present study focuses on empathy, results are only reported for the empathy related measures.

### 2.4. Missing Values

Some participants did not complete all the measures resulting in missing values for either the pretest or the posttest (i.e., 21% to 29%, depending on the examined variable). Missing values frequently occur in applied studies [47], bringing up the question of how to proceed with the incomplete cases. For the study at hand, several data estimation procedures to handle the missing values were considered. In the former analysis of the data by Steffinger [42], listwise deletion (LD) had been chosen, including only the complete cases (i.e., participants who had fully completed the pretest and the posttest questionnaires). However, this procedure results in a reduction of the sample size leading to a loss of statistical power and information [47]. Furthermore, LD requires that missing values are missing completely at random (MCAR, [48]), that is, the fact that a certain value is missing is assumed to be completely independent of all other variables, which is a relatively strong assumption.

As van Ginkel and Kroonenberg [47] note, missing data in a repeated measures design may occur typically due to attrition and is thus usually handled by conducting a multilevel analysis with full information likelihood (FIML) (as described for instance in Hox [49]). Since the model used for the multilevel analysis does not assume equal numbers of observations, the respondents’ missing values do not cause a problem for the analysis [49]. Because missing data in longitudinal research may result in the form of so-called panel attrition (i.e., participants drop out after the first measurement), it can, however, not be considered to be missing completely at random, but rather to be missing at random (MAR, [48]) because some individuals are more likely to drop out than others [49]. In this case, the lack of data depends on other variables in the model. Therefore, the FIML-method is limited as no variables outside the multilevel model can be included in the analysis. Assuming the condition of MAR, the fact that data is missing may depend on auxiliary variables, which are not included in the model. Thus, unbiased results for the analysis cannot be guaranteed [47]. To ensure proper handling of the data, Little’s MCAR test [50] testing against the assumption that the data is MCAR was employed, but showed no significant result (*χ^2^*(12) = 2.75, *p* = 0.997), indicating that the data of this study is in fact not missing in a systematic way but rather (completely) at random (MCAR/MAR).

An imputation-based way of handling missing values is the expectation–maximization (EM) method. As described in Lüdtke, Robitzsch, Trautwein, and Köller [51], it consists of two steps. First, a value is imputed through an iterative procedure, replacing the missing value. Second, parameters are estimated from the now-complete data, which again are included to impute the missing values until the most likely value is reached. Considering the moderate degree of methodological complexity, the EM-algorithm provided the best feasible method to handle the missing values in the present study and was consequently applied on our data set. However, the EM-algorithm’s implementation in the statistical software SPSS is limited to produce only unbiased point estimates (means, variance, covariance), but no standard errors. Thus, analyses using the EM-imputed values may be biased [52] and should be interpreted with caution.

### 2.5. Statistical Analysis

A repeated measures analysis of variance (ANOVA) was employed to analyze the data generated by the present study design. The participants (*N =* 57) were tested in a 2 × 2 factorial design, with a treatment group of size *n =* 35 and a control group of *n =* 22. The condition (treatment group vs. control group) served as the independent variable. The dependent variables were the mean scores from pretest and posttest data of the employed empathy measures (CEEQ, IRI/SPF-E). The repeated measures ANOVA was conducted for the two scales, *Emotional empathy* and *Cognitive empathy*, of the CEEQ [44] and its subscales *Mirroring*, *Empathic Concern*, *Mental State Perception*, and *Perspective Taking* as well as for the subscale *Empathic Concern* of the IRI/SPF-E [46]. Data were analyzed following intention to treat. All statistical analyses were conducted using SPSS 22, with the alpha level set at 0.05.

## 3. Results

Means and standard deviations for each main outcome variable were calculated (see Table 2). The treatment and the control group showed no significant differences at baseline for any of the main outcomes measured. This suggests that participants had been successfully divided at random into the treatment and the control group. Regarding the necessary statistical conditions for the employment of ANOVA, the Levene-tests yielded non-significant results, suggesting homogeneity of variance of the all measured outcome variables. Furthermore, graphic analysis of the outcome variables showed no substantial deviation from the normal distribution.

### 3.1. Descriptive Statistics

Table 2 displays each main outcome variable at pretest and posttest along with the values for the *t*-test at baseline. For the treatment group, the mean values increased from pretest to posttest on every scale and their respective subscales. For the control group, the mean values increased on the *Empathy* (IRI/SPF-E) and *Emotional Empathy* (CEEQ) scales, including the subscales *Mirroring* and *Empathic Concern*, but not for *Cognitive Empathy* and its subscales *Mental State Perception* and *Perspective Taking*.

### 3.2. Inferential Statistics

A repeated measures ANOVA was conducted on the mean scores of each main outcome variables including *Empathy* according to the IRI/SPF-E and *Empathy* according to the subscales of the CEEQ *Emotional Empathy* and *Cognitive Empathy* along with their corresponding subscales (*Mirroring* and *Empathic Concern*; *Mental State Perception* and *Perspective Taking*). *Group* (treatment, control) was the between-subject factor and *time* (pretest and posttest) was the within-subject factor. *F*, *p* and partial *η^2^*-values for the main effects of *time* and the interaction effects of *time × group* for each main outcome are reported in Table 3.

The main effect of *time* yielded significant results for *Emotional Empathy* (*F*(1,55) = 12.55, *p* = 0.001, *η^2^* = 0.19) and its subscales *Mirroring* (*F*(1,55) = 9.22, *p* = 0.004, *η^2^* = 0.14) and *Empathic Concern* (*F*(1,55) = 4.99, *p* = 0.030, *η^2^* = 0.08). For all of the other scales, the main effect of time did not yield statistically significant results. There was no statistically significant interaction effect for the factors *group* and *time* for any of the scales.

## 4. Discussion

This study tested the effects of a DMT intervention on empathy in individuals with ASD. Compared to the wait-list control group, the treatment group showed no substantial increase in any of the employed scales (CEEQ, IRI/SPF-E) after a 10 weeks DMT intervention. Considering the employed procedure to handle the missing data, however, results should be interpreted with caution (see [52] and Section 2.4).

In the prior feasibility study [1], empathy was the only main outcome variable tested that presented no significant changes between treatment group and control group. The present study aimed to refine these findings, by providing results from data collected in a randomized controlled trial which employed an appropriate sample size, that is *N* = 57. Unlike prior attempts to analyze these data which had excluded all cases containing missing values, this study aimed to include those participants through the application of a data estimation procedure for missing values. This procedure, even though improving sample size, may potentially have caused biased results throughout the statistical analyses as discussed above. Regardless of the type of analysis, however, neither the feasibility study [1], nor the study at hand could definitively confirm the hypothesized effects of DMT on the level of empathy in adults with ASD.

In contrast to the feasibility study, where most participants mirrored with a non-autistic partner, participants of the RCT also partnered with each other for the mirroring intervention, which may have caused the difference in results (see elaboration below). In line with our findings, an unpublished Master’s thesis on this RCT [53] reported no changes in self-reported “embodied intersubjectivity”, including correlates of nonverbal empathy measured with the Embodied Intersubjectivity scale (EIS) [18] and discussed potential effects of setting and dose. Kelbel had included the schizophrenic patients from the TESIS study (see [54]), and reported an increase of embodied intersubjectivity that remained significant at follow-up [53].

Koehne and colleagues [37] report an increase in inferred emotions with adults diagnosed with ASD, but corresponding to our findings also not in empathy (i.e., empathic feelings) in their study. The authors discuss whether the construct of empathy may be closer to a state than a trait and thus less stable whereas emotion inference could be connected to a learning process allowing them to apply this capacity in different situations. Participants in our study could have experienced the quality of mirroring during the activity but still not reported this in the assessment at the end as this may be connected to a learning process that requires more time than provided by short-term therapy with a limited number of sessions.

Koehne and colleagues [37] also started to think about mechanisms of change and suggested that imitation and imitation inhibition may have helped develop top-down processes for sociocognitive awareness including emotion recognition and perspective taking through the observation and selection of movement material to imitate. At the same time, they posit that interpersonal movement synchronization influences the participants’ socioaffective experiences of sharing with the other. Manders [55] found that the role of synchronization in mirroring was more nuanced for at least some participants in the current study as increased synchronization was only correlated with increased observable socioaffective connection with the partner under circumstances such as when there were other concurrent interactive behaviors and perhaps when there was more variation between imitation and imitation inhibition. This may point to the need for additional conditions such as sociocognitive attention to the interaction in order for interpersonal movement synchronization to have an effect as a mechanism of change. Investigating mechanisms is a crucial step to improve the focus of the design of both future experiments and DMT interventions.

Participants in the present study received 10 sessions of DMT, offering some time for changes to evolve. Still, this number of sessions can be considered a short-term therapy, and dose effects have so far hardly been discussed with respect to DMT (see e.g., [56]). Anticipating a potential time lag between therapy and observable changes in participants’ behavior, a follow-up was conducted to assess possible long-term changes. Due to the substantial drop out rate of participants, the number of follow-up questionnaires was too low to be analyzed or allow for conclusions on long-term effects. Thus, further studies including a bigger sample should employ a follow-up design, providing longitudinal data with more than two measurement points. Also, the frequency of the intervention could be increased to two or three times a week for ten weeks, to see whether the higher number and frequency of sessions would cause a difference in outcomes.

Nevertheless, the present study addressed an important research gap by providing results on DMT targeting empathy in young adults with ASD. Previous research on therapeutic interventions for this type of patient population mainly focused on children [25]. Moreover, to the authors knowledge there is only a few other studies [1,37,55] assessing empathy as the primary outcome of a DMT intervention, thus the study at hand contributes further evidence on the effects of DMT for individuals with ASD.

### Limitations

The present study has several limitations. First of all, the DMT intervention in itself. In contrast to the feasibility study [1], which paired healthy cotherapists (mostly students of psychology) with participants with ASD for the dyadic mirroring exercise, the present study worked mainly with dyads of individuals with ASD, mirroring each other. Thus, an important reason for the non-significant results of this study could have been the fact that having two socially challenged individuals mirror each other contained more limited, disconnected and less empathically responsive interaction patterns than when individuals with and without ASD mirrored each other. This may have resulted in the development of less body-based empathy and consequently all other forms of empathy, emotional and cognitive empathy. Yet, the relation between these forms of empathy need to first be better understood in general.

It is possible that some participants would have needed more specific guidance on the mirroring interventions from either the therapists or the cotherapists to be able to work on emotions through movement. We assume that autism in both partners in the dyadic mirroring task, could have been an important factor that generally limited the emergence of emotional qualities in the movement and beneficial effects of this task. It seems necessary for one person of the dyad to model empathically responsive movement [36] to create change in the participant. Given that mirroring as a therapeutic method and intervention requires substantial capacities in the person who is mirroring (cf. [31,57]), its quality may have varied and may therefore have resulted in stronger or weaker effects with varying degrees of empathically responsive mirroring. Working constantly with one skilled therapist may thus enforce the mirroring experience and benefit individuals with ASD.

Another limiting factor could have been the reduced sample size, with the percentage of missing data in the range between 21% to 29%, depending on the examined variable. This rate indicates that the handling of missing values is crucial for this study. If only data from participants who had completed questionnaires at both the beginning and at the end of the intervention was taken into account, the sample size would have decreased by enough to preclude certain types of statistical analyses due to insufficient power. This problem was addressed by using the EM-algorithm as a method to estimate the missing values, providing a completed dataset that allowed to run the repeated measures ANOVA for the greater sample. Alternatively, the use of multiple imputation can be considered as an appropriate way to handle the missing values. This procedure is more sophisticated because it provides multiple estimates for the missing values and thus requires additional processing to obtain a pooled result from the data analysis (e.g., [51]). Because of its feasibility and easier handling, the EM algorithm was chosen for the final analysis, but there may exist more appropriate ways to estimate the data. A more sophisticated procedure could eventually result in more accurately estimated data.

Even though the occurrence of missing data is not unusual in applied studies [47], it may still be important to investigate why a substantial number of participants did not complete the questionnaires at both testing times. The drop-out rate was higher for the participants of the control group than for those receiving actual treatment. As the control group did not participate in any alternative activities, they may have experienced a serious decrease in motivation preventing them to participate at both testing sessions. Participants of the control group received the same amount of money after the first round of testing as participants of the treatment group. They were then offered the same number of DMT sessions after the post-test, which may have been too late to keep their interest in the study. Future work could address this type of problem by offering participants in the control condition treatment other than DMT. This placebo treatment could be based on simply dancing for fun or on another physical activity, enabling a better differentiation between constituents of the effect of DMT in comparison to other types of treatment. Further studies could take this into account and could include more than two groups into the study design (e.g., DMT treatment, dance treatment, control condition).

Another critical point is the imbalance in size of the treatment group compared to the control group. This may have affected the validity of the data analysis employed, as this analysis usually requires balanced group sizes (cf. [58]).

The process of randomization can be considered successful as the employed *t*-test when testing at baseline revealed no significant differences between the treatment and the control group. Nonetheless, the treatment group showed lower values for each of the means of the primary outcomes, except for one of the subscales of the CEEQ [44]. This could have hindered the detection of additional significant effects.

All measures analyzed in this study were based on self-reports as the data were collected using questionnaires. However, neither the CEEQ [44] nor the IRI/SPF-E [46] was designed to specifically assess empathy in individuals with ASD. Therefore, it is possible that some participants had difficulty understanding items in the questionnaires. This is supported by some written comments added to the answers given in the questionnaires (e.g., “*Some questions or sentences were not formulated well. I did not know which answer to check*”, “*Several comprehension problems*”). At a more simplistic level, it may be that there were in fact changes in empathy for the individuals, but no appropriate accommodations were made that allowed them to more directly and clearly express them. Instruments adapted to ASD participants’ specific needs may provide more accurate and complete information on their self-perceived empathy [36] and could also be a way to address the difficulties of comprehension described above.

Even self-report measures adapted to the population could be challenging given that individuals with autism may have difficulties identifying their own emotions or have other communication or sensory difficulties that could hinder their understanding or accurate assessment of the items on the measure [7]. To make more accurate assessments, future studies could use a combination of self-report measures, observations, and more flexible interviews in order to better assess the participants’ experiences.

An important question to evaluate is whether the assessment of empathy actually included all aspects of empathy commonly subsumed under this construct. Alongside the two aspects of cognitive and emotional empathy, a third one referred to as motor empathy is supposed to be impaired for individuals with ASD, based on earlier studies [21]. The concept of embodied intersubjectivity (e.g., [18,59]), which appears to be correlated with motor empathy, may also play an important role for a differentiation of the outcomes. Including the aspect of motor empathy, Koch (2015) started to differentiate, which aspects of empathy are most affected by DMT interventions [34].

Furthermore, the CEEQ [44] was employed as a measure to assess empathy, even though a psychometric evaluation only existed for the English version. Though carefully translated, this instrument was not used much by the time the study was conducted.

The inclusion criteria for this study were relatively open, comprising the totality of ASD. Given the diverse symptomatology and resulting impairments, this could have led to varied effectiveness of the treatment for different participants and it is possible that the present intervention did not fit or help all participants (cf. [25]). This included a relatively broad age range of participants (14–52 years), with potential differences in the structural and functional brain development of the young participants (e.g., [60]). Furthermore, this range could even be more problematic as suggested by findings of Hua and colleagues [61] which revealed alterations in the maturation of those brain regions of individuals with ASD related to the main impairments of ASD up to adolescence. Thus, it could be important for further studies to identify subgroups and adapt the chosen interventions more closely to the participants’ needs.

Another limitation as well as an asset could have been the multicenter character of the study. The therapists, cotherapists, locations, and consequently the therapeutic setting itself differed from one site to another. This may, on the one hand, have resulted in higher external validity and increased the plausibility for generalization of the reported findings. On the other hand, the possibility of keeping the experimental conditions standardized, and thus controlling for undesirable external influences, was reduced by the multicenter character of the study.

Lastly, this study focused exclusively on empathy as one of the central impairments in ASD [19], while it may be that other variables such as well-being or other social skills may show a higher potential for change through DMT. In general, more research is needed about the effects of DMT on the various symptoms experienced by individuals with ASD.

## 5. Conclusions

The results of the present experimental study, using a randomized controlled trial design, cannot report conclusive effects in support of its hypothesis on an effect of DMT on empathy for individuals with ASD. Even though the data were collected in a randomized controlled trial, several factors may have limited the outcome of this study including missing data and the use of self-report measures.

In terms of the intervention, we have learned from this study that having the participants mirror among themselves does not improve empathy as tested by this study. Thus, future studies could give more attention to the role of relationship and should test settings in which either the therapist or a cotherapist is doing the mirroring with the participant. This may improve the outcomes on empathy but also the outcomes on other variables such as self-reported well-being, body awareness, self vs. other distinction and social competence as shown in the feasibility study [1]. Also the focus needs to be increasingly on investigating mechanisms of such changes. Mirroring interventions, for example, may have relational and attachment benefits, as suggested by Feniger-Schaal and Lotan [62], and these in turn may have effects on psychological health. The measures used to assess such changes must be selected with the challenges of ASD in mind, such that they include not only self-report measures but also observations and interviews which may allow participants with sensory, motor, or communication challenges to better identify and express their internal emotional experiences (see [8]). 

Further research should thus focus on matching interventions to individual participants with ASD, given the variety of symptoms and presentations of these symptoms included in ASD (cf. [25]). This also includes more specific measures for empathy [36], which ideally should be based on easy-to-understand self-reports and assessment by family and friends as well as unacquainted others. DMT has been shown to be an effective intervention for different clinical populations such as patients with depression, schizophrenia, anxiety, eating disorder, chronic pain, Parkinson, and dementia, affecting several psychological outcomes such as quality of life, well-being, depression, and anxiety (cf. [63]). Besides empathy, including motor empathy (cf. [21]), there are a range of other variables relevant to the impairments of individuals with ASD, which may be sensitive to change through DMT. These other variables should be systematically investigated in future studies, to further shed light on the benefits of DMT for individuals with ASD.

## Figures and Tables

**Table 1 behavsci-08-00061-t001:** Demographic data of the participants for the whole sample and for each group separately.

		Whole Sample ^a^		Treatment ^b^		Control ^c^	
		*n*	%	*n*	%	*n*	%
Treatment location	Bruchsal	6	10.5	6	17.1	0	0.0
	Ludwigshafen	28	49.1	16	45.7	12	54.5
	Karlsruhe	19	33.4	12	34.3	7	31.8
	Missing	4	7.0	1	2.9	3	13.7
Gender	Female	12	21.1	8	22.9	4	18.3
	Male	44	77.2	27	77.1	17	77.3
	Missing	1	1.7	0	0.0	1	4.5
Nationality	German	54	94.6	35	100	19	86.4
	Other	2	3.6	0	0.0	2	9.0
	Missing	1	1.8	0	0.0	1	4.6
Marital status	Not Married	47	82.5	29	82.9	18	81.8
	Married	1	1.8	0	0.0	1	4.5
	Widowed	1	1.8	1	2.9	0	0.0
	Missing	8	13.9	5	14.2	3	13.7
Relationship status	Partner	4	7.0	2	5.7	2	9.1
	No Partner	45	78.9	29	82.9	16	72.7
	Missing	8	14.1	4	11.4	4	18.2
Children	Children	1	1.8	0	0.0	1	4.5
	No Children	46	80.7	30	85.7	16	72.7
	Missing	10	17.5	5	14.3	5	22.8
Medication	Medication	14	26.4	9	25.7	5	22.7
	No Medication	20	35.1	11	31.4	9	40.9
	Missing	23	38.5	15	42.9	8	36.4
Comorbid psychiatric disorders	Other psychiatric diagnosis	15	26.3	8	22.9	7	31.8
	No other psych. diagnosis	20	35.1	13	37.1	7	31.8
	Missing	22	38.1	14	40.0	8	36.4
Clinical status	In-patient	1	1.8	1	2.9	0	0.0
	Out-patient	4	5.0	2	5.8	2	9.1
	Day program	10	17.5	5	14.3	5	22.7
	Missing	42	75.7	27	77.0	15	68.2
Other therapies ^d^	Other therapies	15	26.3	12	34.3	3	13.6
	No other therapies	19	33.3	8	22.9	11	50.0
	Missing	23	40.4	15	42.8	8	36.4

Note. ^a^
*N* = 57; ^b^
*n* = 35; ^c^
*n* = 22; ^d^ mostly physiotherapy, occupational therapy, social skills training, and psychotherapy.

**Table 2 behavsci-08-00061-t002:** Means and standard deviations of the measured main outcome variables and results of the *t*-test for independent samples at baseline.

		Pretest		Posttest		EG and CG at Baseline ^d,e^	
Scale	Group ^a^	*M_pre_*	*SD_pre_*	*M_post_*	*SD_post_*	*t*	*p_baseline_*
**Empathy (IRI/SPF-E)**	EG	3.19	1.02	3.26	0.95	0.42	0.693
	CG	3.33	1.12	3.37	1.11		
**Emotional Empathy (CEEQ) ^b^**	EG	2.06	0.75	2.16	0.69	0.14	0.892
	CG	2.09	0.81	2.28	0.80		
Mirroring	EG	1.81	0.81	1.87	0.73	−0.08	0.935
	CG	1.79	0.91	2.02	0.90		
Empathic Concern	EG	2.34	0.85	2.49	0.77	0.36	0.724
	CG	2.43	0.88	2.57	0.79		
**Cognitive Empathy (CEEQ) ^c^**	EG	2.13	0.60	2.22	0.69	1.58	0.121
	CG	2.38	0.58	2.34	0.68		
Mental State Perception	EG	2.07	0.67	2.17	0.82	1.92	0.060
	CG	2.42	0.67	2.40	0.71		
Perspective Taking	EG	2.20	0.59	2.28	0.75	0.88	0.383
	CG	2.35	0.64	2.28	0.80		

Note. ^a^ EG: treatment group, CG: control group; ^b^ EG: *n* = 35, CG: *n* = 21; ^c^ EG: *n* = 35, CG: *n* = 22; ^d^ IRI/SPF-E: *df* = 54; ^e^ CEEQ: *df* = 55.

**Table 3 behavsci-08-00061-t003:** Results of the repeated measures ANOVA of all measured main outcomes of empathy.

	Time			Time × Group		
Scale	*F*	*p*	*η^2^*	*F*	*p*	*η^2^*
**Empathy (IRI/SPF-E) ^a^**	0.43	0.513	0.01	0.03	0.863	0.00
**Emotional Empathy (CEEQ) ^b^**	12.55	0.001 **	0.19	1.21	0.276	0.02
Mirroring	9.22	0.004 **	0.14	2.97	0.090	0.05
Empathic Concern	4.99	0.030 *	0.08	0.00	0.973	0.00
**Cognitive Empathy (CEEQ)**	0.18	0.671	0.00	1.51	0.255	0.03
Mental State Perception **^b^**	0.44	0.509	0.01	1.00	0.323	0.02
Perspective Taking	0.00	0.966	0.00	0.86	0.358	0.02

*Note. η^2^*: partial eta-squared; ^a^ EG: *n* = 35, CG: *n* = 21; ^b^ EG: *n* = 35, CG: *n* = 22; * *p* < 0.05, ** *p* < 0.01.

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
