# Peer review of "Effects of Dance Movement Therapy on Adult Patients with Autism Spectrum Disorder: A Randomized Controlled Trial"

_behavsci, 2018, doi:10.3390/bs8070061_

Round 1

Reviewer 1 Report

Thank you for this interesting contribution. The article addresses the challenge to define and research specific DMT interventions for specific behaviours of interest, here the development of empathy in individuals with ASD. The study and the conceptual considerations are presented in detail already. The comments in the attached file mainly cover points for further clarification here and there.

Author Response

BehavSci 299721 – Response to Reviewer 1 Comments

Thank you for your comments and questions on our article! Changes in the text are highlighted blue and have a comment that refers to the original line you commented.

-55: “Deficits in social cognition seem to constitute one of the central impairments in ASD (e.g. [5])”: what are causes of ASD and what are consequences of the disorder?

We addressed this in the added paragraph from line 55-65.

-84: The “construct of empathy” is explained -> the conceptualizations of “motor empathy” and “kinesthetic empathy” are meant to be explained by simulation theory/MNS functioning? Some further clarification needed here, also on the consequences for ASD’s.

Please see reformulated and added sentences from line 96 – 109.

-100: For the example box 1: have any delayed responses been considered here?

We have addressed this important point in a further sentence at the end of the case description. Please see line 127-129.

-107 & 112: Check for duplicates, e.g. 107 & 112 could be left out.

112 was dropped

-116: are the interventions also aiming at diminishing repetitive behaviours?

Yes, this was inserted. Since DMT always focuses on authentic expression, this is the case and was inserted. “This includes work on the body level to increase self-recognition, self-other distinction, self-awareness, social interaction and communication, and to decrease repetitive and dysfunctional behavior (e.g. self-harming behavior), taking into account the close relation of physical well-being, emotions, and cognition (increasing the emotional awareness and the ability to language the experience)”

-117: clarification needed: what is “inconsistent responsiveness to interventions” Please see reformulated sentence from line 121 -124.

-124: clarification needed: “including the work and reflection on the non-verbal level and their interaction with expressions on this level”

Our attempt to clarify in the following sentence (line 148-153):

“including work and reflection on the non-verbal level with opportunities to explore sensory and motor experiences and develop bodily awareness and expressivity in movement. Given the potential of sensory and motor challenges to impact individuals’ availability to engage socially, the facilitation of multiple ways of connecting to sensory and motor experiences within the context of an interaction in DMT, may contribute to its therapeutic success in treating individuals with ASD.“

-126: clarification needed: “These may contribute….” -> what are “these”

Please see reformulated sentence from line 148-153.

-131 – 133: there is an assumption presented here that being mirrored will enhance MNS activity – is there any evidence for that?

Two References (Gallese et al. 1996 [28] and Winkielman, McIntosh and Oberman, 2009 [29]) were included in the text.

-138: have vocalizations been included in the intervention?

Yes, they always are. They are supportive, but not well researched.

-127-146 -> all applies to neuro-typical development – any suggestions for ASD’s? mirroring cf. “Generally, ..”

Eberhard-Kaechele follows an analytic developmental concept on stages in interpersonal relating – how would this relate to ASD?

See inserted sentence: “…provides a detailed taxonomy of different mirroring modalities, developing from simple to more complex, which can be used to categorize the developmental stage of mirroring in movement of a client. Koch and colleagues applied this concept in their feasibility study in order to observe and assess the developmental states of the participants over the time of the study. Following this notion, only in the very last modality, which develops between the ages of four and six, the capability of attributing mental states to oneself and others (ToM) is reached with certainty.“

-156: how is Adler’s work connected to the research presented in this paragraph?

It shows anecdotal evidence from two case-studies; as a professionally taped film it is instructive on the use and potential of DMT.

Sentence was added:Adler’s classic video “Looking for me” depicts the process of DMT with a three- and a five-year-old girl with ASD, illustrating the use and potential of DMT.“

-174: are these results from the same data-set as in the study presented here?

Yes. From the clinical observations of the SANS scale by the medical doctors and psychologists involved with the study.

-185: “synchronization and spontaneous mirroring” are new elements here – to which studies would these be related?

The sentence has been re-formulated: “reinforcing their existing positive resources in the domains of synchronization and spontaneous mirroring (e.g. Oberman, Winkielmann, and Ramachandran [40]).”

-205: (M=22.5)?/Ages ranged from 14 to 52 years (M =22.5; SD = 8.52), with the mode at 17 years.?

Mode was dropped, see line 241.

-248: 4 (co-)therapists per group then?

There were two co-therapists per group, either students majoring in dance movement therapy or in psychology. See also line 286.

-257: with no verbal instructions on the movement activities?

Oh, yes, verbal interventions are still important in the movement intervention phase. Sentence therefore has been changed as follows: “Only this last part included substantial verbal intervention, whereas the first three parts were based on movement interventions.”

-273: this is about verbal instruction? -> would the ASD participants have an understanding of what “empathetic mirroring” means?

Please see added sentences in line 310-313 and 317-319.

280, 303: check for use of time

Done.

-291: participants were also meant to mirror the emotional content of the movement?

Yes, this was an attempt to keep participants from mechanically copying merely the shape/form of the movement, which some did in an almost robot like way (very peripheral, without involving core movements), and encourage them to relate on the level of movement qualities (more related to the emotions and an understanding empathic relationship).

Did they have to identify this from the movement?

– No. The groups were too big to do this. We only mentioned (instructively) in the beginning of the second part, when they looked for a partner “And don’t forget that it is not so important to do exactly what the other person does regarding the copying of the outer form, but that this is rather about coming into the same movement qualities to be able to relate to the other”.

-> also, it would be helpful here to provide an explanation whether the conceptual considerations are modelled for ASD specifically or rather stem from frameworks for interventions with neurotypical participants;

Both. The mirroring exercise (or mirror game) is a widely used tool in dance movement and drama therapy with quite a lot of (also attachment-related) research. It has been tested for ASD-participants in our feasibility study (Koch et al., 2015), and worked stunningly well; ASD-participants had a lot of resources re: mirroring and therefore started to enjoy the therapy from the very beginning.

The Baum-Kreis was taken from the work with higher functioning patients with complex trauma; in an attempt to adapt it to ASD participants, we noticed that they profited a lot from it, since (a) they had no social shame to perform in front of the entire group and on the contrary enjoyed it, (b) they practiced socially relevant behavior (as pointed out by their psychologist “this helps them go out to a discotheque”), and (c) some were able to go to the metaphorical level and expressed the content of the lyrics (or the characteristics of the music) in an embodied way. We saw a completely non-verbal participant symbolize to a sad song, showing us that he was able to understand (and express) a lot more than the treatment team thought he could….

We added the blue text “In all groups, the therapist or a co-therapist modeled this way of authentic self-expression to a piece of music in the very first session, followed in movement by all of the participants and the therapist.” (line 327).

-302 -307: how are these body- or movement related aspects related to the verbal processing? Can the relation between verbal processing and movement related changes be clarified here or optional in the discussion? As you refer to the link between movement experience and MNS activity on the one hand and social cognition on the other hand in your introduction, it would be helpful for the reader to know which activities were movement actions/processing (and thus referring to MNS -> your conceptualization of empathy) or which activities were rather verbal actions/processing (referring to social cognition –> mentalization & ToM).

Our attempt to specifiy: See paragraph from line 338 -350.

„...in order to increase awareness of any outcomes experienced at the motor level and bring them to the cognitive level of processing. The co-therapists modeled the expression of emotions and thoughts about their experiences to facilitate the participants’ reflection processes. The verbal processing and feedback after the movement intervention was intended to strengthen body awareness, self-awareness, and self-other awareness as well as empathy, social skills and the connection of sensorimotor and emotional-cognitive levels of experiencing and reflecting. This was anticipated to help participants connect their experiences of motor empathy during the earlier parts of the session to a more general understanding of the other’s perspective and experiences. Participants also completed brief questionnaires on their experience of the session and their connection with their partner, which may have further supported this process.“

-310: check for plural verb/predicate forms after “data…”

Done.

-311: “primary outcomes” = outcome at pre-test?

“primary outcomes” refers to all measures/questionnaires which were used, for clearer understanding sentence was shortened

-315-345: measurement instruments validated for ASD population? -> good that you address this subject in the discussion! Would help the reader to better follow this section if you would provide an argumentation on the choice for non-ASD-specific measurement instruments.

We added the paragraph: Regarding the validation of the questionnaires with ASD participants, the CEEQ was constructed with clinical populations (including ASD) in mind, but is too new to have been validated with the population. It, however seems intuitively suited, because of its many nonverbal components, including “mirroring”. The IRI / SPF-E has been used with samples of the autism spectrum , but we employed only the “Empathic Concern” subscale. The whole battery of questionnaires also included the Boundary Orientation between Self and Other Scale (GO), the Body Self-Efficacy Scale (BSE), and the Embodied Intersubjectivity Scale (EIS) which were developed by our group particularly for the populations of autism and schizophrenia, and were pretested in our feasibility study . As the present study focuses on empathy, results are only reported for the empathy related measures.

-347-389: the diverse methods to correct for missing data is discussed at length, but which one was used for this study?

The EM-algorithm was used, as now clarified in the sentence; „...was the EM-algorithm provided the best feasible method to handle the missing values in the present study and was consequently applied on our data set.“

Results are presented only for MCAR/MAR?

Yes, results are presented under the assuption of MCAR/MAR.

-General remarks: Throughout the article it would be helpful if interventions would be more specifically discussed for their movement related aspects (body/space/effort). Also in the discussion the movement specific effects are not put forward. In this perspective, it would be helpful to have the items of the EIS for example in an additional appendix.

EIS was put into Appendix 1.

-For the discussion: maybe also consider the broad range of age of participants (14 – 52/mode 17yrs.) – as quite a group still is in a developmental phase where social (brain) development is not yet completed (adolescence) (e.g. Blakemore 2018) – and taking into account that for young people with autism there may also occur delays in socio-neurological development/brain maturation, this might even broaden the range (e.g. Hua et al., 2013).

We tried to address this by adding the sentence from line 673-679. The references suggested were also included.

-The discussion could also address the fact that the presented intervention can be considered as modelled (prosocial) behaviours in contrast with spontaneous prosocial behaviours (e.g. line: 274; 300 – 303). As the pro-social behaviours applied stem from neuro-typical individuals this may have influenced the results. Regarding the focus on connection between being mirrored and developing empathy, it would be helpful, if this would be framed by any studies on DMT confirming this connection in neuro-typicals and/or ASD’s. The conceptualization of body-based empathy might be related to this, but is not explained here in terms of behavioural or experiential markers (528-538; 597-601).

-The potential effect of missing data is discussed thoroughly and with clarity. The potential effect of test materials that have not been developed and/or validated for the ASD population is discussed thoroughly for its relevance.

Reviewer 2 Report

As a practitioner in the field who relies on supports that include body movement, body centered activities, rhythm and relationships, I was excited to see the topic of this research and to review the article.  I have seen a wide variety of benefits to the individuals I work with, including positive impacts on the individuals who support/connect with them.  I share that only because I think that is one way in which this research, should they be able to repeat it, could be significantly enhanced...exploring changes in both groups and in relationships.  But, I am ahead of myself.

As excited as I was to hear more about the study, even knowing from the abstract that they had not been able to determine a positive effect, I was disappointed with the limited review and perspective on autism as presented in the introduction.  Since at least as early as 1995 (Donnellan & Leary) and 1998 (Teitelbaum), there has been consideration of at least some aspects of autism as a sensory and movement disorder.  That notion has been supported by both follow up scientific research and first person accounts of the experience of autism.  The work of Chester & Calhoun (2012) suggests that autism begins as a movement disorder at the deepest level, and that it is this difference that then leads to the symptoms that have been used to describe and identify autism.  Most recently, the work of Elizabeth Torres (2017) examines the spectrum of autism within the framework of kinesthetic reafference to help us better understand the different social manifestations we see across the spectrumThe body of these works has lead us to the understanding that for a significant number of autistic individuals there is a  disconnect between what they are feeling or what they know, and their ability to express that. 

The notions posited in this research have been supported by first person accounts of a great number of individuals identified as autistic.  Robledo, Donnellan and Strandt-Conroy (2012) review and explore sensory and movement differences from a first person account.  The data they present help to describe a broader understanding of autism that incorporates the possibility that it is a disorder that affects motor planning, behavior, communication, the sensory motor system, and the dynamic interaction of all of these.  Accounts from individuals like Sue Rubin, Temple Grandin, Daniel Tammet, Jamie Burke, Tracy Thresher, Naoki Higashida, John Elder Robisn, and many others describe how they dynamic systems of movement, cognition and social interactions impact their lives. 

An understanding of the works described above seems necessary to better understand the results of this work.  I would suggest reviewing this body of work and then reconsidering the conclusions based on the results of this study as well as steps for future research.  At it’s most simplistic level, it may be that there were in fact changes in empathy for the individuals, but no accommodations were made that allowed them to more directly and clearly express them.  Understanding this perspective certainly supports some of the changes the authors are already intuiting about how to change the treatment (offering a more inclusive and neurologically diverse group) and giving more attention to the role of relationship and that I would encourage.  

Author Response

Coverletter: BehavSci 299721 – Review 2

Thank you for your comments on our article! Changes in the text are highlighted blue and have a comment to which review they refer.

As a practitioner in the field who relies on supports that include body movement, body centered activities, rhythm and relationships, I was excited to see the topic of this research and to review the article. I have seen a wide variety of benefits to the individuals I work with, including positive impacts on the individuals who support/connect with them. I share that only because I think that is one way in which this research, should they be able to repeat it, could be significantly enhanced...exploring changes in both groups and in relationships. But, I am ahead of myself.

As excited as I was to hear more about the study, even knowing from the abstract that they had not been able to determine a positive effect, I was disappointed with the limited review and perspective on autism as presented in the introduction. Since at least as early as 1995 (Donnellan & Leary) and 1998 (Teitelbaum), there has been consideration of at least some aspects of autism as a sensory and movement disorder. That notion has been supported by both follow up scientific research and first person accounts of the experience of autism. The work of Chester & Calhoun (2012) suggests that autism begins as a movement disorder at the deepest level, and that it is this difference that then leads to the symptoms that have been used to describe and identify autism. Most recently, the work of Elizabeth Torres (2017) examines the spectrum of autism within the framework of kinesthetic re-afference to help us better understand the different social manifestations we see across the spectrumThe body of these works has lead us to the understanding that for a significant number of autistic individuals there is a disconnect between what they are feeling or what they know, and their ability to express that. 

The notions posited in this research have been supported by first person accounts of a great number of individuals identified as autistic. Robledo, Donnellan and Strandt-Conroy (2012) review and explore sensory and movement differences from a first person account. The data they present help to describe a broader understanding of autism that incorporates the possibility that it is a disorder that affects motor planning, behavior, communication, the sensory motor system, and the dynamic interaction of all of these. Accounts from individuals like Sue Rubin, Temple Grandin, Daniel Tammet, Jamie Burke, Tracy Thresher, Naoki Higashida, John Elder Robins, and many others describe how they dynamic systems of movement, cognition and social interactions impact their lives. 

We tried to address this by adding the paragraph from line 56-65, including references [6,7,8] and sentences from line 148-154, including reference [27].

An understanding of the works described above seems necessary to better understand the results of this work. I would suggest reviewing this body of work and then reconsidering the conclusions based on the results of this study as well as steps for future research. At its most simplistic level, it may be that there were in fact changes in empathy for the individuals, but no accommodations were made that allowed them to more directly and clearly express them. Understanding this perspective certainly supports some of the changes the authors are already intuiting about how to change the treatment (offering a more inclusive and neurologically diverse group) and giving more attention to the role of relationship and that I would encourage.

We addressed the issue of the use of better adapted measures/steps for future research in the paragraph from line 653-659, line 707, line 715-720, .We have included your argument in blue directly into the discussion, see line 647-649.

Round 2

Reviewer 2 Report

I appreciated that they incorporated the feedback they were given in the way they did.